# Effective Food Ingredients for Fatty Liver: Soy Protein β-Conglycinin and Fish Oil

**DOI:** 10.3390/ijms19124107

**Published:** 2018-12-18

**Authors:** Tomomi Yamazaki, Dongyang Li, Reina Ikaga

**Affiliations:** 1Department of Nutritional Science, National Institute of Health and Nutrition, National Institutes of Biomedical Innovation, Health and Nutrition, 1-23-1 Toyama, Shinjuku-ku, Tokyo 162-8636, Japan; g1670612@edu.cc.ocha.ac.jp (D.L.); reina017@nibiohn.go.jp (R.I.); 2The Graduate School of Humanities and Sciences, Ochanomizu University, 2-1-1 Otsuka, Bunkyo-ku, Tokyo 112-8610, Japan

**Keywords:** β-conglycinin, fish oil, NAFLD, alcohol-induced fatty liver, PPARγ2

## Abstract

Obesity is prevalent in modern society because of a lifestyle consisting of high dietary fat and sucrose consumption combined with little exercise. Among the consequences of obesity are the emerging epidemics of hepatic steatosis and nonalcoholic fatty liver disease (NAFLD). Sterol regulatory element-binding protein-1c (SREBP-1c) is a transcription factor that stimulates gene expression related to de novo lipogenesis in the liver. In response to a high-fat diet, the expression of peroxisome proliferator-activated receptor (PPAR) γ2, another nuclear receptor, is increased, which leads to the development of NAFLD. β-Conglycinin, a soy protein, prevents NAFLD induced by diets high in sucrose/fructose or fat by decreasing the expression and function of these nuclear receptors. β-Conglycinin also improves NAFLD via the same mechanism as for prevention. Fish oil contains n-3 polyunsaturated fatty acids such as eicosapentaenoic acid and docosahexaenoic acid. Fish oil is more effective at preventing NAFLD induced by sucrose/fructose because SREBP-1c activity is inhibited. However, the effect of fish oil on NAFLD induced by fat is controversial because fish oil further increases PPARγ2 expression, depending upon the experimental conditions. Alcohol intake also causes an alcoholic fatty liver, which is induced by increased SREBP-1c and PPARγ2 expression and decreased PPARα expression. β-Conglycinin and fish oil are effective at preventing alcoholic fatty liver because β-conglycinin decreases the function of SREBP-1c and PPARγ2, and fish oil decreases the function of SREBP-1c and increases that of PPARα.

## 1. Introduction

In Western societies, the prevalence of obesity, which is associated with increased fat or carbohydrate consumption, has increased dramatically [1,2,3]. Obesity clinically leads to nonalcoholic fatty liver disease (NAFLD) [4], type 2 diabetes [5,6,7], and coronary heart disease [8]. NAFLD is defined by excessive hepatic triglyceride (TG) content in the absence of excessive alcohol consumption. NAFLD itself is also a potential risk factor for the development of both type 2 diabetes and metabolic syndrome [9,10,11]. Moreover, NAFLD is a precursor of the more advanced liver disease nonalcoholic steatohepatitis (NASH), a condition that may progress to cirrhosis in up to 25% of patients [12,13].

Two major causes of NAFLD in humans are overconsumption of sucrose/fructose and fat. Using stable isotopes, it was shown that in healthy men, a high-fructose diet increased fractional de novo lipogenesis six-fold [14]. For fatty liver induced by a high-fat (HF) diet, patients with NAFLD have been shown to consume significantly more saturated fat per day compared with control subjects who were matched for body mass index [15]. The amount of hepatic fat appears to be related to the amount of fat in the diet rather than to endogenous fat deposits in obese women [16], which suggests that increased postprandial TG may favor fatty liver formation.

The plasma non-esterified fatty acid pool contributes most of the fatty acids that flow to the liver, and the fatty acids are secreted by the liver in very low-density lipoprotein (VLDL) particles [17]. The quantifiable biological sources of hepatic TG were directly detected in NAFLD patients by Donnelly et al. [18], as follows: 59.0 ± 9.9% of TG were from non-esterified fatty acids, which flow into the liver via the lipolysis pathway in adipose tissue; 26.1 ± 6.7% were from de novo lipogenesis; and 14.9 ± 7.0% were from dietary fat. Sterol regulatory element-binding protein-1c (SREBP-1c) is a transcription factor that stimulates the expression of genes related to de novo lipogenesis [19,20]. Additionally, SREBP-1c mRNA and protein levels and TG levels increase in the liver under HF diet conditions [21]. Peroxisome proliferator-activated receptor (PPAR) γ2 is another nuclear receptor that is involved in lipid metabolism in metabolic tissues [22]. PPARγ2 expression is greatly increased in response to an HF diet, especially a diet high in saturated fatty acids rather than unsaturated fatty acids, and it can lead to the development of NAFLD [23]. Fatty acid metabolism-related gene expression in NAFLD (*n* = 26) and normal liver (*n* = 10) samples, which were obtained by liver biopsy, was evaluated and SREBP-1c, PPARγ, and their target gene expression in NAFLD livers were higher compared with the normal liver [24]. Another study enrolling 22 NAFLD patients and 16 control subjects also revealed that liver PPARγ and SREBP-1c mRNA expression was higher in NAFLD patients compared with control subjects [25]. Moreover, immunohistochemical results showed that the expression of adipose differentiation-related protein, which is a target gene of PPARγ and localized on the surface of lipid droplets in hepatocytes, was increased in the hepatocytes in patients with a fatty liver compared with the normal liver [26]. However, clinical studies from NASH patients using PPARγ agonists have not detected a worsening of steatosis [27,28]. These discrepancies should be considered.

Besides NAFLD, the earliest stage of alcohol-induced liver disease is called alcoholic fatty liver. Moreover, excessive alcohol consumption may progress to hepatitis and fibrosis, which may lead to liver cirrhosis. Alcohol-induced liver cirrhosis was responsible for 493,300 deaths worldwide in 2010, representing 0.9% of all global deaths and 47.9% of all liver cirrhosis deaths [29]. After acute or chronic ethanol consumption, hepatic TG levels are increased through dysregulation of lipid metabolism, and several studies indicated that alcohol exposure stimulates lipogenesis, such as via SREBP-1c and PPARγ, and inhibits fatty acid oxidation in the liver [30,31,32,33,34].

Japan has the largest aging societies because of longevity [35]. The Japan Public Health Center-based prospective study showed that typical dietary habits in Japan include a high consumption of soy/isoflavones, fish/n-3 polyunsaturated fatty acids (PUFAs), salt/salted foods, and green tea, and a low consumption of red meat and saturated fat; there was also inverse associations between soy/isoflavones or fish/n-3 PUFAs and diabetes [36]. In this review, we focus upon the effects of β-conglycinin, a soy protein, on the prevention and improvement of NAFLD. Soybean is a popular ingredient that is used in Japanese foods such as tofu, miso, and natto. We also refer to the effects of fish oil on the prevention of NAFLD. Fish oil contains PUFAs, eicosapentaenoic acid (EPA), and docosahexaenoic acid (DHA). We also discuss the effects of β-conglycinin and fish oil on alcoholic fatty liver.

## 2. Effects of β-Conglycinin

Soybeans (*Glycine max* L.) provide one of the most abundant plant sources of dietary protein. The protein content of soybeans varies from 36% to 56%, and this diversity results from the areas in which the soybeans are grown [37,38]. Eighty percent of soybean protein is made up of glycinin (11S globulin) and β-conglycinin (7S globulin) [38]. β-Conglycinin accounts for 30% of soy protein, which is the second-most abundant soy protein component [39]. β-Conglycinin is a trimeric protein that is composed of the subunits α (approximately 67 kDa), α′ (approximately 71 kDa), and β (approximately 50 kDa) [40]. All subunits are *N*-glycosylated, and α and α′ have extension regions that, in addition to the core regions, are common to all subunits [40].

### 2.1. Preventive Effects of β-Conglycinin on Nonalcoholic Fatty Liver Disease (NAFLD)

β-Conglycinin effectively prevented fatty liver induced by an HF diet [41]. In this study, β-conglycinin or casein (20 energy% (en%) protein) was administered to low-safflower oil (10 en% fat)-fed, high-safflower oil (60 en% fat)-fed, and high-butter (60 en% fat)-fed mice. After 11 weeks of feeding, the HF dietary groups of mice fed casein as a protein source showed higher body weights and liver TG concentrations compared with the control mice fed a low-fat diet containing casein. Among the dietary groups, mice fed a high-butter diet containing casein showed the most profound liver TG accumulation. However, liver TG concentrations of mice fed an HF (high-safflower oil and high-butter) diet containing β-conglycinin were markedly decreased. The mechanisms underlying the prevention of NAFLD in mice supplemented with β-conglycinin are described below. HF diets increased PPARγ2 mRNA expression relative to the control diet. mRNA of fatty acid translocase (CD36), a target gene of PPARγ2, was also increased. However, β-conglycinin administration effectively suppressed the expression of PPARγ2 and its target genes. There were no significant differences in mRNA levels of PPARα, which is responsible for fatty acid oxidation, and its target genes (medium-chain acyl-CoA dehydrogenase (MCAD), acyl-CoA oxidase, and carnitine palmitoyltransferase (CPT) 1) between the β-conglycinin-supplemented and β-conglycinin-non-supplemented groups. The concentration of serum adiponectin secreted from adipose tissue after feeding an HF diet for at 11 weeks was higher in β-conglycinin-supplemented mice compared with β-conglycinin-non-supplemented mice (*p* < 0.05). β-Conglycinin supplementation in rats was also reported to increase adiponectin [41,42,43]. Adiponectin plays an important role in glucose and lipid metabolism and obesity. This adipocytokine activates muscle glucose utilization but also induces muscle and hepatic fatty acid oxidation, decreases hepatic glucose production, and decreases circulating TG and free fatty acid (FFA) concentrations [44,45,46]. A strong negative correlation between plasma adiponectin levels and body mass indices were also reported [47]. β-Conglycinin causes a significant decrease in the rat plasma TG concentration via a reduction in the VLDL-TG concentration [42]. Although increased circulating plasma adiponectin may explain beneficial physiological responses that are driven by dietary β-conglycinin, the molecular mechanism for increasing adiponectin gene expression remains unknown. Serum insulin concentrations after 11 weeks of feeding were lower in the β-conglycinin-supplemented mice than in the β-conglycinin-non-supplemented mice (*p* < 0.05), and β-conglycinin supplementation in rats was also reported to increase insulin sensitivity compared with no β-conglycinin supplementation [41,42]. β-Conglycinin also increases fibroblast growth factor 21 (FGF21) gene expression in the liver and circulating FGF21 levels [48]. FGF21 induces lipolysis and glucose uptake through the activation of hormone-sensitive lipase and glucose transporter 4, respectively [49,50]. Thus, β-conglycinin has beneficial effects on health, at least partially through these circulating mediators, although the mechanism of how β-conglycinin regulates these mediators remains unknown.

Next, β-conglycinin was tested to prevent NAFLD induced by diets high in sucrose [41]. Sucrose supplementation for 11 weeks caused NAFLD in mice and increased mRNA expression of the transcription factors SREBP-1c and carbohydrate response element-binding protein (ChREBP) in the liver. SREBP-1c regulates regulatory enzymes acetyl-CoA carboxylase 1 (ACC1), fatty acid synthase (FAS), and stearoyl-CoA desaturase 1 (SCD1) expression, leading to fatty liver [51]. ChREBP binds to and activates the transcription of several lipogenic enzyme genes such as liver-type pyruvate kinase (LPK). β-Conglycinin significantly prevented an increase in mRNA expression of SREBP-1c and ChREBP, and lipogenic genes such as FAS, SCD1, ACC1, and LPK in sucrose-supplemented mice. Thus, preventing an increase in PPARγ2 mRNA expression in HF-fed mice and in SREBP-1c and ChREBP mRNA expression in sucrose-supplemented mice, which were observed in β-conglycinin administration, contributed to the anti-NAFLD effects of β-conglycinin. It is conceivable that a reduction in liver PPARγ2 protein may be a result of hormonal changes (increased adiponectin concentration and decreased basal insulin concentration) via reducing adipose tissue, rather than a direct effect of β-conglycinin or its metabolite to decrease liver PPARγ2 mRNA expression.

### 2.2. Improvement Effects of β-Conglycinin on NAFLD in Diet-induced Obese Mice

Dietary β-conglycinin also improves NAFLD in HF diet-induced obese (DIO) mice [52]. The DIO mice, which were fed an HF diet for 4 weeks to generate diet-induced obesity, were grouped into the HF, medium-fat (30 en%), and low-fat (10 en%) groups. Additionally, casein or β-conglycinin was given as a dietary protein to the DIO mice, resulting in a total of six groups. After feeding the experimental diets for 5 weeks, liver TG accumulation was greater with higher fat in the diet, and regardless of the dietary fat level, the β-conglycinin-supplemented mice had significantly lower TG accumulation. Moreover, the leptin and insulin concentrations were lower in the β-conglycinin-fed group than in the casein-fed group. However, β-conglycinin had no effect on the concentration of serum adiponectin in this case. Hepatic PPARγ2 and CD36 mRNA expression increased in the β-conglycinin-non-supplemented HF diet-fed mice, and β-conglycinin significantly reduced PPARγ2 and CD36 mRNA expression. The amount of nuclear PPARγ2 protein was also significantly lower in the β-conglycinin-supplemented HF diet-fed mice compared with the β-conglycinin-non-supplemented HF diet-fed mice. β-conglycinin significantly suppressed mRNA expression of SREBP-1c and its target genes, such as FAS, SCD1, and ACC1. Conversely, PPARα and its target gene, MCAD, were not affected by β-conglycinin supplementation. An increase in fatty acid oxidation might not be necessary for β-conglycinin to decrease the hepatic TG concentration. These data indicate that the suppression of PPARγ2 and SREBP-1c and their target genes may contribute to the mechanisms causing the improvement of HF diet-induced NAFLD. Insulin plays a critical role in lipogenesis through activating SREBP-1c expression and activity [19], and a lower concentration of serum insulin was observed in the β-conglycinin-supplemented mice. However, the mechanism for insulin to increase SREBP-1c mRNA has been not clarified. Protein kinase C beta is reported to be a key mediator of insulin-mediated activation of hepatic SREBP-1c and its target lipogenic genes [53]. Additionally, two transcription factors, liver X receptor (LXR) α and C/EBP (CCAAT/enhancer-binding protein) β, are necessary but not sufficient for insulin induction of hepatic SREBP-1c mRNA [54]. Recently, basic helix-loop-helix family member E40 was reported to be a third transcription factor that is required for insulin induction of SREBP-1c mRNA expression in liver, but it is not sufficient to account for the increase in SREBP-1c mRNA expression caused by insulin [55]. An additional event seems to be required. Moreover, decreased PPARγ2 protein in the liver by knockdown of PPARγ2 mRNA reduced the expression of its target genes and de novo lipogenesis-related genes even though there were no alterations in SREBP-1c mRNA expression [23]. Although it remains unclear whether PPARγ2 regulates the transcription of de novo lipogenesis-related genes directly or indirectly, β-conglycinin seems to decrease the mRNA expression of these genes at least partially through the decrease of PPARγ2 expression.

### 2.3. Preventive Effects of β-Conglycinin on Alcohol-Induced Fatty Liver

β-conglycinin supplementation also protected against alcohol-induced fatty liver and NAFLD [56]. Before ethanol administration, mice were fed 20 en% β-conglycinin or casein for 4 weeks. After feeding these diets, mice were administered ethanol or glucose, as a control, by gavage. There was no difference in hepatic TG concentration between β-conglycinin and casein supplementation when glucose was administered. However, when ethanol was administered, hepatic TG was increased in the casein- and β-conglycinin-fed mice, but this increase was significantly lower in the β-conglycinin-fed mice compared with the casein-fed mice. Although the SREBP-1c expression was not affected by alcohol administration, expression of the target genes FAS, ACC1, and SCD1 was increased. β-Conglycinin supplementation did not affect SREBP-1c mRNA expression and the amount of mature SREBP-1 protein. However, β-conglycinin decreased the basal (without ethanol) mRNA expression of SREBP-1c and its target genes, such as FAS, SCD1, and ACC1, indicating that β-conglycinin suppressed the basal expression level of these genes. When ethanol was administered, β-conglycinin supplementation significantly suppressed SCD1 expression to the levels of mice fed casein without ethanol. The ethanol-administered casein-fed mice showed slightly increased PPARγ2 mRNA expression compared with the glucose-administered casein-fed mice and the amount of nuclear PPARγ2 protein was significantly increased, whereas an increase of this protein in the mice fed β-conglycinin was not caused by ethanol administration. Ethanol significantly increased CD36 mRNA expression in the mice fed casein, but not in the mice fed β-conglycinin. PPARα mRNA expression decreased in response to ethanol in both the casein- and β-conglycinin-fed mice.

### 2.4. How Does β-Conglycinin Work?

In humans, the estimated average intake of β-conglycinin is very low (less than 1 en%). The mean intake of soybeans and related foods in Japanese people was 58.6 g/day, and their protein intake was 5.1 g [57]. Because 30% of the soy protein was β-conglycinin, about 1.5 g of β-conglycinin was consumed per day on average [39]. If the average energy intake in humans is assumed to be 2000 kcal/day and energy from protein is 4 kcal/g, then 1.5 g of β-conglycinin corresponds to 0.3 en%. In humans, additional supplementation of 5 g of β-conglycinin was reported to effectively reduce intra-abdominal obesity [58]. This amount corresponds to 1 en%; this dose is lower than 20 en%, which is the dose used in the study mentioned above [38], which was required to elicit a lower white adipose tissue weight in ddY mice, suggesting that humans might be more sensitive to β-conglycinin than ddY mice. Although there is concern that β-conglycinin is an allergen, there was no mention of allergy in these reports [58,59].

The peptide generated from β-conglycinin seems to play a role in lipid metabolism, although this mechanism has not been identified. Small peptides released by β-conglycinin digestion were shown to activate low-density lipoprotein receptors and decrease lipoprotein-containing apolipoprotein B-100 secretion in cultured HepG2 cells [60,61]. KNPQLR, EITPEKNPQLR, and RKQEEDEDEEQQRE, three peptides that are in purified β-conglycinin hydrolysates, show FAS-inhibitory biological activity [62]. Thus, some peptides released from β-conglycinin digestion may directly or indirectly affect lipid metabolism in adipose tissue and the liver.

β-Conglycinin is also involved in the prevention and improvement of hyperlipidemia and obesity. Administration of β-conglycinin to rats prevented the increase in serum total cholesterol, TG, and very low-density lipoprotein-TG [42,63,64]. Moreover, in a human randomized, double-blind, placebo-controlled study, consumption of 5 g of β-conglycinin per day for 12 weeks significantly reduced serum TG concentrations in subjects with hypertriglyceridemia from 2.65 to 2.29 mmol/L, whereas consumption of 5 g of casein did not have this effect [58]. As stated above, β-conglycinin increases serum adiponectin and FGF21 levels and decreases insulin levels, which leads to the prevention or improvement of NAFLD, although the mechanism is unclear. Thus, β-conglycinin may be a promising dietary protein for the amelioration of NAFLD and obesity.

## 3. Effects of Fish Oil

Fish oil contains n-3 fatty acids such as EPA and DHA. Fish synthesize these n-3 PUFAs from ingesting marine plants. Fish that is rich in n-3 PUFAs are those that store lipids in their muscle tissue, and leaner fish containing lower n-3 PUFA levels are those that store lipids in their liver [65]. n-3 PUFAs are ligands for PPARα and they can modulate lipid metabolism by enhancing fatty acid β-oxidation and decreasing de novo lipogenesis [66]. When hypertriglycemic patients ate these PUFAs as part of their diet, their blood TG concentrations decreased such that PUFAs are considered to have protective effects against fatty liver [67]. These effects were reported to mainly result from the combined effects of inhibiting lipogenesis and stimulating fatty acid oxidation in the liver [68,69].

### 3.1. Preventive Effects of Fish Oil on NAFLD Induced by a High-Fat Diet

EPA or DHA supplementation accelerates chylomicron TG clearance by increasing lipoprotein lipase activity, and EPA and DHA appear to be equally effective [70]. The atherogenic HF (AHF) diet is used as a dietary model of NASH, which progresses from NAFLD in mice. While both NAFLD and NASH are characterized by hepatosteatosis, NASH is characterized by hepatic damage, inflammation, oxidative stress, and fibrosis [71]. When C57BL/6J mice were fed the AHF diet, including HF and high-sucrose and supplementation with or without 5% EPA ethyl ester or DHA ethyl ester, for 4 weeks, EPA had a greater effect on reducing liver TG levels compared with DHA. Conversely, DHA had a greater suppressive effect compared with EPA on AHF diet-induced hepatic inflammation and reactive oxygen species generation [72]. C57BL/6J mice were fed an HF diet (60 en%) for 5 months, with the fat in the form of safflower oil or fish oil, and compared with the safflower oil-fed mice, the fish oil-fed mice showed a decreased TG concentration in liver [68]. Using low-density lipoprotein receptor-knockout mice as a preclinical model of Western diet-induced NASH, it was reported that DHA attenuated hepatic inflammation, oxidative stress, and fibrosis [73,74]. Thus, although both EPA and DHA are n-3 PUFAs, their function in the amelioration of NAFLD and NASH seems to be slightly different. EPA, but not DHA, were also reported to be the hypotriglyceridemic components of fish oil, and mitochondria, but not peroxisomes, were their principal target [75]. DHA can inhibit TG secretion from intestinal epithelial cells via PPARα activation [76]. Additionally, microRNAs (miRNAs) are small non-protein-coding RNAs that bind to specific mRNAs and inhibit translation or promote mRNA degradation. The fish oil supplementation also altered hepatic expressions of miRNAs, such as rno-miR-29c, rno-miR-328, rno-miR-30d, and rno-miR-34a, which may contribute to fish oil’s amelioration of NAFLD in Sprague Dawley rats [77].

Feeding a diet that was free from sucrose, and in which one part of the fat was replaced by fish oil, to the ddY strain of mice showed that the effects of dietary fish oil on NAFLD differed according to the etiology of the fatty liver, as follows: fish oil prevents sucrose-induced fatty liver but exacerbates HF-induced fatty liver [78]. The ddY mice displayed marked postprandial hypertriglyceridemia in response to dietary fat [79]. To examine the effects of fish oil in an HF diet including 60 en% fat, 10 en% fat was replaced with 10 en% fish oil. The liver TG concentration in HF-fed mice was higher than that in control mice, but fish oil did not affect the HF-induced increase in liver TG at all [78]. Fish oil further increased PPARγ1, PPARγ2, and CD36 mRNA expression that was increased via an HF diet. Fish oil did not alter the PPARα mRNA level, but it increased acyl-CoA oxidase, CPT1, and MCAD mRNA expression, suggesting that PPARα activation may not be sufficient to decrease the hepatic TG accumulation in these mice. When C57BL/6J mice were fed HF diets including various amounts of fish oil, two mechanisms that reduced amount of mature, active SREBP-1 protein through fish oil feeding were reported: inhibition of the SREBP-1 proteolytic cascade when the mice were fed low amounts of fish oil (10–30 en%); and inhibition of the SREBP-1 proteolytic cascade and a decrease of SREBP-1 mRNA expression when the mice were fed high amounts of fish oil (about 50 en%), but the hepatic TG concentration was not reported [69]. In this previous study [78], SREBP-1c mRNA levels were reduced by fish oil when the mice were fed an HF diet especially a diet that was high in saturated fatty acid. However, the hepatic TG content was not decreased. Therefore, the exacerbation of NAFLD may be because of a further increase in PPARγ expression. It is likely that fish oil supplementation is less effective in preventing fatty liver in mice that are overfed with fat.

There are several reasons why the effects of fish oil are different among these reports. The first reason is the differing mouse strains among the reports. The second reason is the differing HF-diet composition among the reports, which may or may not include sucrose as part of the diet. The third reason is the differing amount of fish oil among the studies. Fish oil has preventive effects against development of sucrose-induced fatty liver, as described below. The fish oil effects on HF diet-induced fatty liver seem to depend on experimental conditions. However, humans obtain various types of nutrients from various foods. Thus, it is important to clarify the mechanism, but as mentioned below, a clinical study on the effects of fish oil on fatty liver showed that fish oil seemed to be effective if various types of food are ingested in an appropriate volume and composition.

### 3.2. Preventive Effects of Fish Oil on NAFLD Induced by High-Sucrose Diet

In the study mentioned above, control mice were fed a high-starch diet containing 70 en% starch, and the sucrose-supplemented mice were fed a high-starch diet plus 20% sucrose (*w*/*w*) in the drinking water. Ten en% fish oil was added to the high-starch diet for 11 weeks to examine the effects of fish oil [78]. The liver TG concentration in sucrose-supplemented mice was higher than in the control mice, and fish oil completely prevented the sucrose-induced increase in liver TG. Sucrose-supplementation resulted in a 2.2-fold increase in SREBP-1c, FAS, ACC1, and LPK mRNA expression, but not in SCD1 mRNA expression compared with the high-starch-fed mice. Fish oil significantly decreased FAS, SCD1, and ACC1 mRNA expression, indicating that the inhibition of sucrose-induced SREBP-1c activation may lead to a decrease in the liver TG content. The fish oil (from tuna) used in this study contained 7% EPA and 24% DHA. Other fish oils, such as a mixture of tuna and sardine oils containing 6% EPA and 13% DHA, and sardine oil containing 28% EPA and 12% DHA, were also effective in preventing sucrose-induced hepatic TG accumulation. The suppressive effect of n-3 PUFA on hepatic lipogenic enzyme genes is caused by a decrease in the mature form of SREBP-1 protein, which is presumably a result of reduced cleavage of the SREBP-1 precursor protein [80]. n-3 PUFAs suppress this SREBP-1c gene expression, which is crucial for lipogenesis, by inhibiting the LXR-retinoid X receptor that binds to the LXR-responsive elements [81]. The order of inhibitory magnitude of each long-chain fatty acid on SREBP-1c expression is as follows: arachidonic acid > EPA > DHA > linoleic acid ≫ oleic acid > saturated fatty acid = 0. 

Fructose is also known to stimulate de novo lipogenesis in the liver [82]. Fish oil was effective in fructose-induced NAFLD, and it also inhibited hepatic lipogenesis, increased hepatic β-oxidation, and improved insulin sensitivity [83,84]. DHA alone was reported to protect against fructose-induced hepatocellular lipid accumulation, through alleviating the endoplasmic reticulum stress response [85]. A high-fructose diet increased fractional de novo lipogenesis six-fold and supplementation with fish oil (7.2 g/day) partially prevented this increase [14]. It is likely that most humans are responders to sucrose/fructose overconsumption and that fish oil supplementation inhibits sucrose-induced fatty liver. People in the United States have an average intake of n-3 fatty acid of about 0.7 en% [86], whereas in Eskimos, the average intake of n-3 fatty acid, which is mostly from fish oils, is about 5 en% [87]. Thus, 10 en% fish oil in the diet, which contained 3.1 en% DHA plus EPA in these mouse studies, corresponds to the average fish oil intake in Eskimos.

### 3.3. Fish Oil in Human NAFLD

Several clinical studies have evaluated whether n-3 PUFA fish oil plays a role in treatment of NAFLD. There were 42 patients with NAFLD enrolled into a study, and n-3 PUFA ethyl ester 1-g capsules were ingested daily for 12 months. Sixty-four percent of patients in the n-3 PUFA group showed significantly improved NAFLD, whereas no significant change occurred in the control group [67]. In another study, patients with NAFLD were enrolled and randomized into a double-blind, placebo-controlled trial (DHA + EPA ethyl ester, 4 g/day, *n* = 51; placebo, *n* = 52). Treatment for 15–18 months with n-PUFA decreased liver fat and erythrocyte DHA enrichment was associated with a decrease in the liver fat percentage [88]. The effects of n-3 PUFAs on NASH patients were also investigated. n-3 PUFAs at 3000 mg/day for 1 year did not lead to an improvement in the primary end point of NASH activity score reduction (n-3, *n* = 17; placebo, *n* = 17). However, n-3 PUFAs led to a significant decrease in the liver fat content, as assessed using MRI (magnetic resonance imaging) [89]. An additional prospective, randomized, double-blind placebo-controlled study enrolled 37 NAFLD and NASH patients [90]. Supplementation of n-3 PUFA containing EPA (2160 mg) and DHA (1440 mg) daily for 48 weeks provided no beneficial effects, even on hepatic steatosis. However, a pilot study involving 23 biopsy-proven NASH patients showed promising results of improved hepatic steatosis and fibrosis with EPA (2700 mg/day for 12 months) [91]. Another larger-scale study was also reported [92]. Subjects with NASH and NAFLD were assigned to groups that took a placebo (*n* = 75), or high-dose (2700 mg/d; *n* = 86) or low-dose (1800 mg/d; *n* = 86) EPA ethyl ester for 12 months. EPA ethyl ester had no significant effects on steatosis, inflammation, ballooning, or fibrosis scores. The effects of n-3 PUFAs in obese children were also reported. DHA (250 and 500 mg/day, *n* = 20 in each group; placebo, *n* = 20) was supplemented for children with NAFLD for 6, 12, 18, and 24 months. Both doses of DHA supplementation improved liver steatosis in children with NAFLD [93,94].

Several systematic reviews and meta-analyses on the efficacy of n-3 PUFA supplementation in NAFLD with or without NASH were also reported. A systematic review and meta-analysis of clinical trials assessing the efficacy of dietary n-3 PUFA supplementation showed beneficial changes in liver fat with n-3 PUFA treatment, but n-3 PUFA did not significantly improve liver fibrosis in NAFLD and NASH patients [95,96,97,98]. In children with NAFLD, a systematic review and meta-analysis to evaluate the effectiveness of PUFA supplementation was also performed. A meta-analysis of four randomized clinical trials conducted in 263 children showed that long-term n-3 PUFA supplementation can improve liver steatosis assessed using a liver ultrasound, with no side effects [99].

Thus, the effects of EPA and DHA for the patients with NAFLD and NASH seem to be controversial. If there are some effects, they seem to be limited to a decrease in liver fat, but the optimal dose is currently unknown.

### 3.4. Preventive Effects of Fish Oil on Ethanol-Induced Fatty Liver

Dietary fish oil was shown to be useful in preventing ethanol-induced fatty liver [30]. When mice were fed fish oil (30 en%) 1 day before ethanol administration, ethanol-induced fatty liver was markedly reduced by 73%. There may be multiple mechanisms of ethanol-induced fatty liver as mentioned above: ethanol administration results in the activation of SREBP-1c and ChREBP, which promotes de novo fatty acid synthesis; an increase PPARγ and acyl-CoA:diacylglycerol acyltransferase mRNA expression promotes TG synthesis; and decreases in PPARα and CPT1 mRNA expression, which leads to the inhibition of fatty acid oxidation [29]. Fish oil in the diet decreased SREBP-1c activity and increased PPARα activity and counteracted the ethanol-induced activation of SREBP-1c and inactivation of PPARα.

## 4. Conclusions

Fatty liver is induced through the activation of lipogenesis including SREBP-1c and PPARγ, and the downregulation of fatty acid oxidation in the liver. The effects of dietary components on fatty liver differ based on its etiology. β-Conglycinin is effective at preventing NAFLD caused by overfeeding either an HF or high-sucrose diet (Figure 1), and it is also effective at improving symptoms of NAFLD. Conversely, the effects of dietary fish oil on NAFLD differ based on the etiology of fatty liver. Fish oil is effective at preventing NAFLD induced by sucrose/fructose because SREBP-1c activity is inhibited. However, the effect of fish oil on NAFLD induced by fat is controversial because fish oil further increases PPARγ2 expression depending on the experimental conditions such as the type of HF-diet and whether it includes sucrose, the ratio of included fish oil, and the mouse strain (Figure 1). Fish oil seems to be effective if various kinds of foods are eaten in an appropriate volume and composition. Both β-conglycinin and fish oil are effective in preventing alcohol-induced fatty liver.

## Figures and Tables

**Figure 1 ijms-19-04107-f001:**
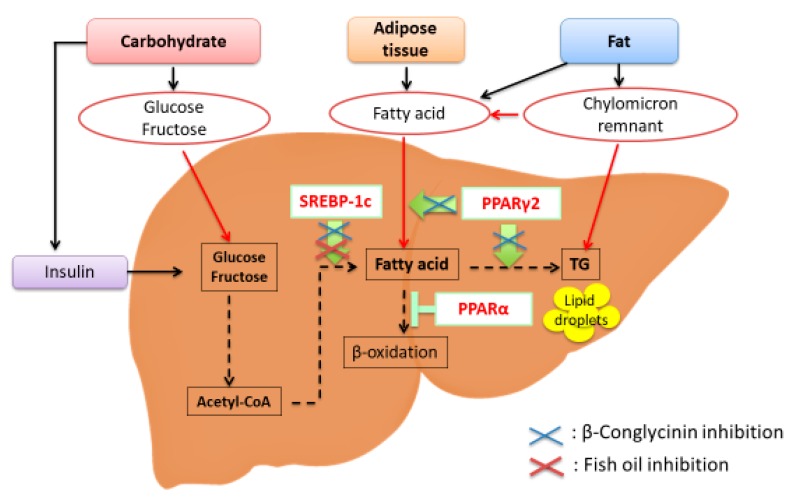
Transcription factors involved in the etiology of nonalcoholic fatty liver disease (NAFLD). Sterol regulatory element-binding protein-1c (SREBP-1c) is a transcription factor that stimulates the expression of de novo lipogenesis-related genes. Peroxisome proliferator-activated receptor γ2 (PPARγ2) is involved in lipid metabolism, and PPARγ2 expression is greatly increased in response to a high-fat diet, especially a diet that is high in saturated fatty acids. PPARα is responsible for fatty acid oxidation. β-Conglycinin downregulates SREBP-1c and PPARγ2 (blue crosses). Fish oil downregulates SREBP-1c (red cross).

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
