# Peer review of "Effective Food Ingredients for Fatty Liver: Soy Protein β-Conglycinin and Fish Oil"

_ijms, 2018, doi:10.3390/ijms19124107_

Round 1

Reviewer 1 Report

This manuscript was reviewed that effective food ingredients for fatty liver: soy protein β-conglycinin and fish oil.

This manuscript has less scientific information overall.

It is already known that β-conglycinin regulated not only transcription factors of lipogenesis but also cytokines (FGF21 and so on).

Authors should describe the preventive mechanism of fatty liver by β-conglycinin more in detail.

Response: We thank the reviewer for comments.

As the reviewer pointed out, β-conglycinin was reported to induce adiponectin and FGF21 and these circulating mediators are thought to bring about beneficial effects on the preventive mechanism of fatty liver by β-conglycinin. So we added the sentences in Lines 140-149, and Lines 153-159 as follows.

“Adiponectin plays an important role in glucose and lipid metabolism and obesity. This adipocytokine activates muscle glucose utilization but also induces muscle and hepatic fatty acid oxidation, decreases hepatic glucose production, and decreases circulating TG and free fatty acid (FFA) concentrations [44-46]. A strong negative correlation between plasma adiponectin levels and body mass indices were also reported [47]. β-Conglycinin causes a significant decrease in the rat plasma TG concentration via a reduction in the VLDL-TG concentration [42]. Although increased circulating plasma adiponectin may explain beneficial physiological responses that are driven by dietary β-conglycinin, the molecular mechanism for increasing adiponectin gene expression remains unknown.”

β-Conglycinin also increases fibroblast growth factor 21 (FGF21) gene expression in the liver and circulating FGF21 levels [48]. FGF21 induces lipolysis and glucose uptake by activation of hormone-sensitive lipase and glucose transporter 4, respectively [49, 50]. Thus, β-conglycinin has beneficial effects on health at least partially through these circulating mediators, although the mechanism of how β-conglycinin regulates these mediators remains unknown.

Authors described that fish oil prevents sucrose-induced fatty liver but exacerbates HF-induced fatty liver.

However, in recently, many papers clarified that fish oil prevents HF-induced fatty liver using miRNA or knockout mice. Therefore, the conclusion by authors is unacceptable.

Response: We thank the reviewer for comments.

We had introduced the results from ddY mice alone. As the reviewer pointed out, preventive effects of fish oil on HF-induced fatty liver were reported, using C57BL/6J mice and knockout mice. Moreover, it was also reported that hepatic expression of miRNAs altered by the fish oil supplementation. So we added the sentences in Lines 309-333, and Lines 360-370 as follows.

“EPA or DHA supplementation accelerates chylomicron TG clearance by increasing lipoprotein lipase activity, and EPA and DHA appear to be equally effective [70]. The atherogenic HF (AHF) diet is used as a dietary model of NASH, which progresse from NAFLD in mice. While both NAFLD and NASH are characterized by hepatosteatosis, NASH is characterized by hepatic damage, inflammation, oxidative stress, and fibrosis [71]. When C57BL/6J mice were fed the AHF diet, including HF and high-sucrose and supplementation with or without 5% EPA ethyl ester or DHA ethyl ester, for 4 weeks, EPA had a greater effect on reducing liver TG levels compared with DHA. Conversely, DHA had a greater suppressive effect compared with EPA on AHF diet-induced hepatic inflammation and reactive oxygen species generation [72]. C57BL/6J mice were fed a HF diet (60 en%) for 5 months, with the fat in the form of safflower oil or fish oil, and compared with the safflower oil-fed mice, the fish oil-fed mice showed a decreased TG concentration in liver [68]. Using low-density lipoprotein receptor-knockout mice as a preclinical model of western diet-induced NASH, it was reported that DHA attenuated hepatic inflammation, oxidative stress, and fibrosis [73, 74]. Thus, although both EPA and DHA are n-3 PUFAs, their function in amelioration of NAFLD and NASH seems to be slightly different. EPA, but not DHA, were also reported to be the hypotriglyceridemic components of fish oil and mitochondria, but not peroxisomes, were their principal target [75]. DHA can inhibit TG secretion from intestinal epithelial cells via PPARα activation [76]. Additionally, microRNAs (miRNAs) are small non-protein-coding RNAs that bind to specific mRNAs and inhibit translation or promote mRNA degradation. The fish oil supplementation also altered hepatic expressions of miRNAs, such as rno-miR-29c, rno-miR-328, rno-miR-30d, and rno-miR-34a, which may contribute to fish oil’s amelioration of NAFLD in Sprague-Dawley rats [77].”

“There are several reasons why the effects of fish oil are different among these reports. The first reason is the differing mouse strains among the reports. The second reason is the differing HF-diet composition among the reports, which may or may not include sucrose as part of the diet. The third reason is the differing amount of fish oil among the studies. Fish oil has preventive effects against development of sucrose-induced fatty liver, as described below. The fish oil effects on HF diet-induced fatty liver seem to depend on experimental conditions. However, humans obtain various types of nutrients from various foods. Thus, it is important to clarify the mechanism, but as mentioned below, a clinical study on the effects of fish oil on fatty liver showed that fish oil seemed to be effective if various types of food are ingested in an appropriate volume and composition.”

Reviewer 2 Report

This review summarizes the literature and current knowledge on the effects of soy protein

Β-Conglycinin and fish oil in the prevention and treatment of fatty liver, both of non alcoholic and alcoholic origin. The text is easy to read and well structured, reviewing the most important articles in the field. I have some comments:

- English must be improved and the text should be proofread by a native speaker. There are plenty of mistakes,

Response: We thank the reviewer for the positive comments on our study. We had native speakers proofread our manuscript and expressed our thanks in Acknowledgments.

- the last paragraph of the Abstract is contradictory: “... Fish oil is more effective in preventing NAFLD induced by sucrose/fructose than by fat because fish oil inhibits increases in the expressions of SREBP-1c and the target genes but not PPARγ2. However, alcohol intake causes alcoholic fatty liver, which is induced by the increased expressions of SREBP-1c and PPARγ2 and the decreased expression of PPARα. β-Conglycinin and fish oil are effective in the prevention of alcoholic fatty liver because they decrease the functions of SREBP-1c and PPARγ2.” ...> either fish oil does not increase PPARγ2 or it does, one of the two.

Response: We apologize for our grate mistake. As the reviewer pointed out, fish oil does not decrease PPARγ2. So we changed these sentences in Lines 32-35 as follows.

“β-Conglycinin and fish oil are effective at preventing alcoholic fatty liver because they decrease the functions of SREBP-1c and PPARγ2 β-conglycinin decreases the function of SREBP-1c and PPARγ2, and fish oil decreases the function of SREBP-1c and increases that of PPARα.”

- the authors should justify why they chose to write a review on these two nutrients. While soybean could make sense because it is integral in the Japanese dietary patterns, the choice of fish oil is less clear. Why these two have been chosen for review and not not other 3 or 4? Perhaps the synergism between the two could be better discussed.

Response: In the Japan Public Health Center-based prospective (JPHC) Study, typical dietary habits in Japan are high consumption of soy/isoflavones, fish/n-3 polyunsaturated fatty acids (PUFAs), salt/salted foods and green tea, and low consumption of red meat and saturated fat (Tsugane, S.; Sawada, N., The JPHC study: design and some findings on the typical Japanese diet. Jpn. J. Clin. Oncol. 2014, 44, (9), 777-82.). Inverse associations between food ingredients and diabetes were only found in soy/isoflavones or fish/n-3 PUFAs in this study. Moreover, our manuscript has been getting longer in consequence of the revision according to the reviewers’ comments. So, we focused on these two food ingredients. We are also interested in the synergism between these two food ingredients but such data are not available for now. So we added the sentences in Lines 92-97 as follows.

“Japan has the largest aging societies because of longevity and the proportion of the elderly in the population [35]. The Japan Public Health Center-based prospective Study showed that typical dietary habits in Japan include a high consumption of soy/isoflavones, fish/n-3 polyunsaturated fatty acids (PUFAs), salt/salted foods, and green tea, and a low consumption of red meat and saturated fat; there was also an inverse associations between soy/isoflavones or fish/n-3 PUFAs and diabetes [36]”.

Round 2

Reviewer 1 Report

Authors response for my questions and comments. 

Reviewer 2 Report

All comments have been addressed.